# Four-Color Pseudovirus-Based Neutralization Assay: A Rapid Method for Evaluating Neutralizing Antibodies Against Quadrivalent Hand, Foot, and Mouth Disease Vaccine

**DOI:** 10.3390/vaccines13030320

**Published:** 2025-03-18

**Authors:** Fan Gao, Lingjie Xu, Qian Wang, Gang Wang, Mingchen Liu, Lu Li, Qian He, Xuanxuan Zhang, Ying Wang, Qunying Mao, Zhenglun Liang, Tao Wang, Xiao Ma, Xing Wu

**Affiliations:** 1School of Life Sciences, Tianjin University, Tianjin 300072, China; gaofan@nifdc.org.cn; 2NHC Key Laboratory of Research on Quality and Standardization of Biotech Products, NMPA Key Laboratory for Quality Research and Evaluation of Biological Products, State Key Laboratory of Drug Regulatory Science, Research Units of Innovative Vaccine Quality Evaluation and Standardization, Chinese Academy of Medical Sciences, National Institutes for Food and Drug Control, Beijing 102629, China; wangqian187@nifdc.org.cn (Q.W.); liumingchen@nifdc.org.cn (M.L.); lilulilu@nifdc.org.cn (L.L.); zhangxx_01@163.com (X.Z.); wangying7586@nifdc.org.cn (Y.W.); maoqunying@nifdc.org.cn (Q.M.); lzhenglun@126.com (Z.L.); maxiao@nifdc.org.cn (X.M.); 3Vazyme Biotech Co., Ltd., Nanjing 210089, China; xulingjie@vazyme.com (L.X.); wanggang03@vazyme.com (G.W.)

**Keywords:** hand, foot, and mouth disease, enterovirus, quadrivalent vaccine, pseudovirus-based neutralization assay

## Abstract

Background/Objectives: Enterovirus 71 (EV71) and coxsackieviruses A16 (CA16), A10 (CA10), and A6 (CA6) are the primary pathogens that cause hand, foot, and mouth disease (HFMD). Currently, many manufacturers are developing bivalent, trivalent, and tetravalent vaccines that target these antigens. Cell-based neutralization assay (CBNA), the gold standard for detecting neutralizing antibodies (NtAbs), which are used as indicators of HFMD vaccine efficacy, has several limitations. We aimed to develop a novel assay for detecting NtAbs against a quadrivalent HFMD vaccine. Methods: We developed a four-color pseudovirus-based neutralization assay (PBNA), utilizing fluorescent reporter genes, to rapidly evaluate neutralizing antibodies against EV71, CA16, CA10, and CA6 in multivalent vaccines and compared it with CBNA. Results: PBNA could rapidly and simultaneously detect NtAbs against the four serotypes and required lesser amounts of sera compared to CBNA. A good consistency in determining NtAb titers was observed for PBNA and CBNA. Conclusions: PBNA provides a robust tool for evaluating the efficacy of multivalent HFMD vaccines and conducting seroepidemiological studies.

## 1. Introduction

Hand, foot, and mouth disease (HFMD) is an acute infectious condition predominantly observed in children under the age of five years, although it can also affect adults [1]. Enteroviruses (EVs), particularly enterovirus 71 (EV71) and coxsackievirus A16 (CA16), are the predominant pathogens responsible for HFMD outbreaks, either individually or concurrently [2]. Since 2010, coxsackievirus A6 (CA6) and coxsackievirus A10 (CA10) have emerged as significant causative agents of HFMD outbreaks and epidemics in Asia, America, and Europe [3,4,5,6,7]. Therefore, HFMD has become a major public health concern.

As the most effective and cost-efficient preventive measure against HFMD, EV71-based vaccines have garnered significant attention in countries of the Asia–Pacific region over the past decade. Since December 2015, three Chinese manufacturers have successively received market authorization for their inactivated whole-virus EV71 vaccines, which have significantly reduced the incidence of severe HFMD cases and related fatalities, providing robust protection for millions of infants and young children [8,9]. However, the successful development of the monovalent EV71 vaccine is only an initial step toward the prevention and control of HFMD [10]. Owing to the absence of cross-protection among different EVs, the monovalent EV71 vaccine alone is insufficient in curbing outbreaks and widespread prevalence of HFMD caused by other EV pathogens, potentially leading to recurrent infections in children. To achieve broader prevention of HFMD and effectively address the threat posed by it, the development of multivalent HFMD vaccines has emerged as a key direction in next-generation vaccine research [10]. Numerous manufacturers worldwide are currently developing bivalent, trivalent, and quadrivalent vaccines against EV71, CA6, CA10, and CA16 [11,12,13,14,15].

Neutralizing antibodies (NtAbs) are critical indicators of HFMD vaccine efficacy. The cell-based neutralization assay (CBNA) is considered a gold standard for detecting NtAbs in serum from individuals vaccinated against HFMD. This method typically requires multiple corresponding authentic viruses in separate experiments to individually detect NtAbs for each serotype [16,17]. However, this approach is time-consuming, demands large quantities of serum, and imposes stringent biosafety requirements owing to the handling of live viruses. To address these challenges, we previously developed neutralization detection methods utilizing luciferase-reporter pseudoviruses. However, these methods cannot simultaneously detect NtAbs for multiple serotypes present in multivalent vaccines [18,19,20]. In this study, we developed a novel four-color pseudovirus-based neutralization assay (PBNA) for EV71, CA6, CA10, and CA16 (Appendix A). This assay offers several advantages: it can simultaneously detect NtAbs against four serotypes, thereby enhancing the detection efficiency; the total detection time is reduced from seven days to two days; serum consumption is minimized to one-fourth of that required for CBNA; and the single-cycle infection model employed in PBNA enhances experimental safety. This innovation is essential for evaluating multivalent HFMD vaccines and for conducting population immunity studies.

## 2. Materials and Methods

### 2.1. Cell and Virus

HEK-293T (293T) and human malignant embryonic rhabdomyosarcoma (RD) cells were obtained from the American Type Culture Collection (ATCC, Manassas, VA, USA). HEK-293F, Vero, and HeLa cells were generously provided by Vazyme Biotech Co., Ltd. (Nanjing, China). All cell lines were maintained in Dulbecco’s modified Eagle medium (DMEM, Gibco, Waltham, MA USA) supplemented with 10% fetal bovine serum (ExCell Bio, Shanghai, China) and 1% penicillin/streptomycin solution (Gibco, USA) at 37 °C in a 5% CO_2_ atmosphere. The four enterovirus strains used for detection, namely EV71 [19], CA16 [21], CA10 [18], and CA6 [20], were obtained from the National Institutes for Food and Drug Control (NIFDC), Beijing, China.

### 2.2. Serum

Serum was obtained by immunizing NIH mice with two doses of the vaccines. Monovalent and quadrivalent vaccines against EV71, CA16, CA10, and CA6 were provided by Minhai Biotechnology Co., Ltd. (Beijing, China). Additionally, 56 serum samples from nonimmunized mice served as negative controls to establish the cut-off values. Anti-poliovirus sera, with an NtAb titer of 1:256 against poliovirus type III, were derived from rats immunized with the inactivated poliomyelitis vaccine, Sabin strains. Anti-hepatitis A virus serum was obtained from mice immunized with an inactivated hepatitis A vaccine that exhibited a total IgG antibody titer greater than 1:10^5^ as determined using ELISA. The sera were donated by Sinovac Biotech Co., Ltd. (Beijing, China). National standard NtAbs against EV71 (NIFDC code: 300017), CA16 (NIFDC code: 300030), CA10 (NIFDC code: 300044), and CA6 (NIFDC code: 300043) were supplied by the NIFDC.

### 2.3. Fluorescent Gene Selection

*BFP* (mTagBFP2, GenBank: XAX95701), *GFP* (EGFP, GenBank: ANC98519), *RFP* (DsRed-Express, GenBank: QPB77308), and *CFP* (E2-Crimson, GenBank: AMO27221) were chosen as test fluorescent reporter genes. Following transfection of the expression plasmids for these four fluorescent reporter genes into 293T cells, the fluorescence signals were detected in specific excitation/emission wavelength channels.

### 2.4. Construction of Pseudovirus Packing Plasmid and Detection of Fluorescent Gene Expression

#### 2.4.1. Genomic Sequences of the Viruses

RNA from four enterovirus standard detection strains (EV71, CA16, CA10, and CA6) provided by the NIFDC was reverse-transcribed into cDNA. Full-length DNA fragments encompassing the viral genomes were amplified using three replicates of overlapping PCR. Reference sequences were obtained via Sanger sequencing. The 5′- and 3′-ends of the viral genomes were amplified and sequenced using 5′/3′-RACE. Finally, the complete genomic sequences of the four enteroviruses were successfully determined.

#### 2.4.2. Vector Backbone

The modified pcDNA3.1(+) fluorescent protein gene expression plasmid was used as the vector backbone for construction of the pseudovirus, and the expression plasmids for enterovirus replicons and capsids were constructed.

#### 2.4.3. Capsids

The P1 sequences encoding capsid proteins of EV71, CA16, CA10, and CA6 were codon optimized (GENERAL BIOL, China) for human cells, cloned into the vector backbone, and expressed under the control of the CMV promoter [22].

#### 2.4.4. Replicons

All the constructs were developed using a CMV promoter system. For the EV71 replicon, the sequence of the EV71 genome was modified by deleting the *P1* gene and replacing it with the *BFP* fluorescent reporter gene followed by the *EV71 2A* peptide gene at the 3′-end. Additionally, a poly(A) sequence (20A), hepatitis D virus ribozyme (HDVRz), and an SV40 poly(A) signal sequence were sequentially incorporated into the vector backbone. The EV71 replicon was packaged with the corresponding capsid to form an EV71 pseudovirus.

For the CA10 replicon, based on a previous report [18] that enterovirus replicons can be heterologously packaged into pseudoviruses of other serotypes, we replaced the *BFP* gene in the EV71 replicon with CFP. The resulting CA10 replicon was successfully heterologously packaged with the CA10 capsid to form a CA10 pseudovirus.

Attempts to heterologously package CA16 using the EV71 replicon system were unsuccessful. Therefore, we constructed a CA16 replicon using the CA16 sequence. This replicon was packaged with the CA16 capsid to form a CA16 pseudovirus, albeit with a relatively low viral titer. Because we speculated that the packaging efficiency of CA16 could be influenced by the length of the viral genome, we modified the length of the CA16 replicon by incorporating the *P1* gene segment and successfully packaged the CA16 pseudovirus.

For the CA6 replicon, we used the CA16 replicon as previously described, and substituted the *GFP* gene with *RFP*. This enabled us to successfully heteropack CA6 pseudoviruses.

#### 2.4.5. Detection of Fluorescence Gene Expression

By transfecting 293T cells with replicons and their corresponding capsids, we successfully generated pseudoviruses expressing four distinct reporter genes. To validate the specific expression of fluorescent genes and assess the potential interference in machine-based fluorescence detection, the four pseudovirus types were analyzed using different excitation/emission wavelength channels.

### 2.5. Pseudovirus Preparation

The capsid and replicon plasmids were individually mixed with Lipomaster 293 Transfection Reagent at a ratio of 1:3 (*w*/*v*). Subsequently, these plasmids were cotransfected into 293F cells at a ratio of 1:1 (*w*/*w*). Following a 48 h transfection period, the supernatant was harvested through a single freeze–thaw cycle (from −80 °C to room temperature) and centrifugation at 4000× *g* for 20 min.

### 2.6. Pseudoviral Titer Estimation

One hundred microliters of RD cell suspension containing 0.5 × 10^6^ cells/mL was added to each well of a 96-well cell culture plate and incubated overnight at 37 °C in a 5% CO_2_ atmosphere. The next day, serial dilutions of the pseudovirus were prepared in a 96-well deep-well plate. The medium in the 96-well cell culture plate was replaced with 100 μL of DMEM supplemented with 2% FBS, after which 100 μL of the diluted pseudovirus from the deep-well plate was added to each well. A cell control (CC) was also established. The plates were then incubated at 37 °C in a 5% CO_2_ atmosphere for 19–21 h. Fluorescence signals were detected using a fluorescent enzyme-linked immunospot analyzer (CTL S6 Universal M2). The pseudoviral titer was calculated as follows:Pseudoviral titer (pfu/μL) = the number of fluorescent spots × dilution factor ÷ 100

### 2.7. Neutralization Test

After inactivation at 56 °C for 30 min, the serum samples were diluted 1:30 with 2% FBS-containing DMEM and subsequently subjected to a two-fold serial dilution in a 96-well deep-well plate. An equal volume of pseudovirus was added to both the diluted serum and viral control (VC). The subsequent procedures were the same as those described in Section 2.6. The infection inhibition rate (%) was calculated as follows:Infection inhibition rate (%) = (VC value − CC value − sample test value) ÷ (VC value − CC value) × 100%

The neutralizing antibody titer (NT_50_) was determined as the reciprocal of the dilution factor that achieved a 50% infection inhibition rate.

### 2.8. Statistical Analysis

All data were analyzed using Microsoft Excel 2010, SPSS Statistics 19, and GraphPad Prism 8.0. Statistically significant differences between two groups were determined using unpaired Student’s *t*-tests. A two-sided *p* < 0.05 was considered to be statistically significant. Spearman’s rank correlation coefficient was used for correlation analysis, and a *p*-value < 0.05 was considered to indicate a statistically significant correlation.

## 3. Results and Discussion

Considering the possibility of cross-interference between different fluorescence signals, we selected GFP, CFP, BFP, and RFP to achieve non-interference in the detection of pseudoviruses [23]. We transfected plasmids expressing BFP, GFP, RFP, and CFP fluorescent proteins, all of which were clearly differentiated using the instrument (Figure 1A). Using distinct packaging strategies, we successfully packaged four pseudoviruses with different fluorescent expression genes (Figure 1B–F). After infecting RD cells, the four-color fluorescent pseudoviruses were specifically detected using the corresponding wavelengths of excitation/emission light (Figure 1G). CMV eukaryotic expression promoters were used to express the capsid proteins and replicons of pseudoviruses, thereby simplifying the process and reducing the costs associated with extracellular transcription of T7 promoters or cotransfection of T7 RNA polymerase [14]. Notably, the CA6 capsid could not package CA6 replicons but could package CA16 replicons to form CA6 pseudoviruses. These findings differ from previously reported results [18,19,20], suggesting that specific interactions among viral capsids, genomes, and reporter genes may play a role, warranting further investigation.

We optimized the method based on six critical factors that influenced the performance of the PBNA: cell line selection for neutralization, testing time point, pseudovirus addition volume, cell inoculation density, cell generation, and incubation duration for neutralization. RD cells were found to be the most susceptible to the EV71, CA16, CA10, and CA6 pseudoviruses (Figure 2A). The optimal testing time point was determined to be between 19 and 21 h post-infection (Figure 2B), with an ideal pseudovirus addition amount of 500 pfu/well (Figure 2C). The optimal cell inoculation density was 4 × 10^4^–6 × 10^4^ cells/well (Figure 2D). Additionally, minimal variability (GCV < 30%) was observed in P3 and P14 (Appendix A), and the incubation duration for optimal neutralization was 2 h (Appendix A).

For method validation, we first conducted a specificity study. First, we used 56 serum samples from unimmunized mice as negative controls to determine the NT_50_. The cut-off values for EV71, CA16, CA10, and CA6 pseudoviruses were 15, 15, 18, and 11, respectively. To ensure robust test results, a conservative cut-off value of 40 was adopted (Figure 2E). Samples with an NT_50_ value < 40 were assigned a value of 20. Second, we used anti-EV71, anti-CA16, anti-CA10, anti-CA6, anti-polio, and anti-HAV sera to evaluate the specificity of PBNA. A weak cross-neutralization reaction (NT_50_ = 241) was observed when detecting the cross-neutralization of EV71 pseudovirus against anti-CA16 serum, whereas the NT_50_ value of the CA16 pseudovirus detected using anti-CA16 serum was 4557 (Figure 2F). This phenomenon may be attributed to the presence of common cross-neutralizing epitopes between EV71 and CA16 [24,25,26]. Previous studies have shown that, in authentic virus cross-neutralization assays, the cross-NtAb titers of anti-CA16 serum detected using the EV71 authentic virus can reach 6.25% [27], whereas our study yielded a titer of 5.29%. However, no cross-neutralization activity was observed between the CA16 pseudovirus and anti-EV71 serum. Although the underlying mechanism remains unclear, weak cross-protection against CA16 and EV71 has been noted in CA16-immune serum, leading us to hypothesize that cross-protection epitopes originate from SCARB2 receptor-binding domains [24,25]. To better assess the immunogenicity of the EV71 vaccine, we introduced single-point mutations (S213I or S213T in P1) into the common neutralizing epitopes of EV71 and CA16; however, this did not reduce the weak cross-reactivity of anti-CA16 serum to EV71. Additionally, antisera against all types, including anti-poliovirus and anti-HAV serum, did not exhibit cross-neutralizing activity against pseudoviruses other than their respective types. Third, we analyzed the interference of the diluent. The sample was diluted two-fold with the specified diluent, and each dilution was independently assayed for NtAbs. A linear relationship was noted between the logarithm of the dilution factor and that of the NT_50_ (R^2^ > 0.98) (Figure 2G).

The precision and accuracy of this method were evaluated using a nested design. The intermediate precision ranged from 11.9% to 20.9% for EV71, from 7.2% to 31.5% for CA16, from 4.8% to 6.7% for CA10, and from 9.0% to 16.2% for CA6 (Appendix A). Accuracy assessments revealed that the absolute relative bias was between 4.8% and 19.6%, 2.7% and 10.1%, 0.9% and 7.3%, and 10.3% and 17.9% for EV71, CA16, CA10, and CA6, respectively (Appendix A).

Neutralizing antibodies were detected using both authentic and pseudoviral particles in sera from 10 mice immunized with the quadrivalent vaccine, and the correlation between the two methods was evaluated. The detection trends for authentic viruses and pseudoviruses were largely consistent across EV71, CA16, CA10, and CA6 (Figure 2H). Spearman correlation analysis revealed *p*-values of 0.007, 0.012, 0.001, and 0.013 for EV71, CA16, CA10, and CA6, respectively, indicating a strong concordance between the two methods. Analysis of the NT_50_ derived from authentic and pseudoviral assays showed that PBNA exhibited a good correlation with the classical neutralization titer method, with pseudoviral titers being higher than those for authentic viruses, suggesting a potentially greater sensitivity of the new method. We hypothesized that this increased sensitivity may be due to differences in the packaging strategies of pseudoviruses compared with those for authentic viruses. Authentic viruses produce procapsids, A-particles, and empty particles during replication [28], which can lead to antibody depletion in neutralization assays. In contrast, pseudoviral packaging, which is influenced by the transfection plasmid affecting *P1* gene expression, allows for high-level replication of cotransmissible replicons, potentially enhancing the ratio of the capsid to replicating gene. However, additional research is required to confirm this hypothesis.

This study had some limitations, notably the absence of testing serum immunized with human vaccines and evaluation of the consistency between PBNA and CNBA for these human-immunized sera. These assessments will be addressed in future clinical trials on quadrivalent HFMD vaccines.

## 4. Conclusions

We developed a novel four-color PBNA, capable of simultaneously detecting NtAbs against EV71, CA16, CA10, and CA6 antigens, based on fluorescent reporter genes. Compared with the traditional CBNA, this four-color PBNA offers significant advantages, including high throughput, reduced cost, rapid detection, minimal sample volume, enhanced safety, simplified operation, and automated data acquisition. This assay provides a robust tool for evaluating the efficacy of multivalent HFMD vaccines and conducting seroepidemiological studies. Additionally, it should facilitate high-throughput screening of antiviral compounds targeting viral entry and replication with lower biosafety requirements.

## Figures and Tables

**Figure 1 vaccines-13-00320-f001:**
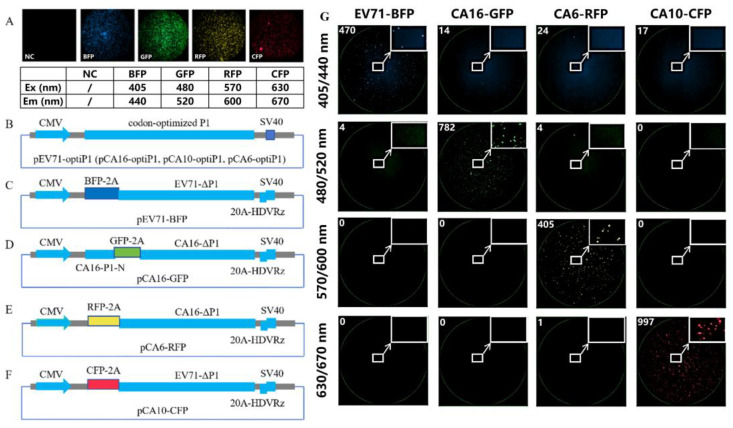
Selection of four-color fluorescent reporter genes and packaging strategies for pseudoviruses. (**A**) Four distinct fluorescent protein expression signals can be detected using the instrumentation. (**B**) Four enterovirus capsid expression plasmids were constructed using a consistent strategy. (**C**) Strategy for constructing the EV71 replicon. (**D**) Strategy for constructing the CA16 replicon. (**E**) Strategy for constructing the CA6 replicon. (**F**) Strategy for constructing the CA10 replicon. (**G**) After infecting RD cells with the four-color fluorescent pseudoviruses, excitation/emission light channels at specific wavelengths can be specifically detected. CMV: Cytomegalovirus promoter element; codon-optimized P1: codon-optimized *P1* gene; SV40: SV40 polyA signal sequence; pEV71-optiP1, pCA16-optiP1, pCA10-optiP1, and pCA6-optiP1: plasmids encoding the capsids of EV71, CA16, CA10, and CA6, respectively; BFP-2A: 3′-end of *BFP* fluorescent protein gene linked to 2A peptide (AITTL); GFP-2A: 3′-end of *GFP* fluorescent protein gene linked to 2A peptide (KITTL); RFP-2A: 3′-end of *RFP* fluorescent protein gene linked to 2A peptide (KITTL); CFP-2A: 3′-end of *CFP* fluorescent protein gene link to 2A peptide (AITTL); EV71-ΔP1: genome sequence of EV71 with the *P1* gene deleted; 20A-HDVRz: 20A-HDVRz: a polyA tail with 20 adenine bases added to the 3′-end of the replicon and a self-cleaving sequence of hammerhead ribozyme; CA16-P1-N: N-terminal 1839 bases of the *P1* gene of CA16; pEV71-BFP, pCA16-GFP, pCA6-RFP, and pCA10-CFP: plasmids encoding the replicons of EV71, CA16, CA6, and CA10, respectively.

**Figure 2 vaccines-13-00320-f002:**
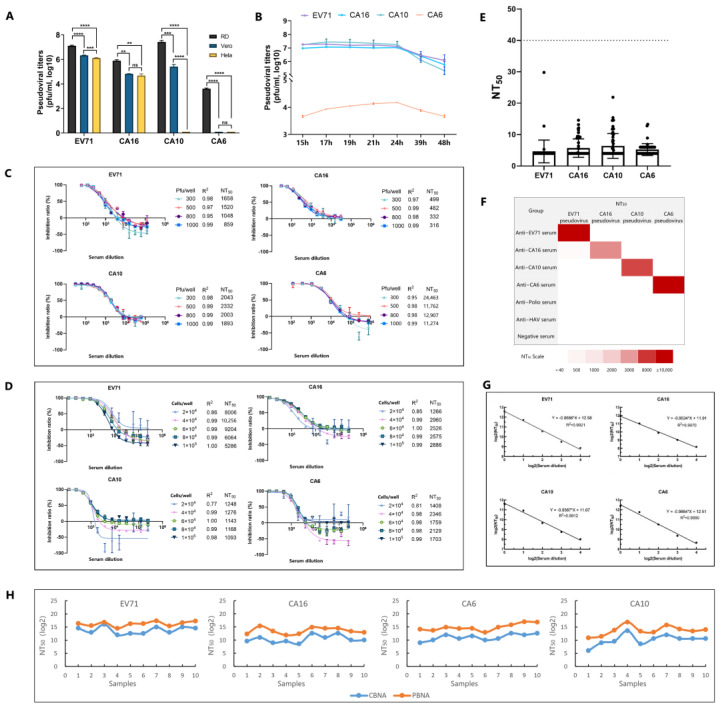
Optimization of four parameters for the pseudovirus-based neutralization assay (PBNA), the specificity as part of method validation, and a correlation analysis of PBNA and cell-based neutralization assay (CBNA) results. (**A**) Cell line selection for neutralization. Negative results were designated as 0.1; (**B**) testing time point; (**C**) pseudovirus addition volume; (**D**) cell inoculation density. (**E**) A total of 56 serum samples from nonimmunized mice served as negative controls to establish the cut-off value using four-color PBNA. Cut-off value = Mean + 3SD. (**F**) Mice or rats were immunized with vaccines against EV71, CA16, CA10, CA6, poliovirus, and HAV to obtain the corresponding antisera. The specificity of the pseudovirus detection results was evaluated using four-color PBNA. (**G**) The logarithmic conversion of NT_50_ values and sample dilutions exhibits excellent linearity, with R^2^ > 0.98. (**H**) Correlation between PBNA and CBNA test results. CBNA: NT_50_ was defined as the highest dilution capable of inhibiting 50% of the cytopathic effect. PBNA: NT_50_ was defined as the reciprocal of the dilution factor that achieved 50% infection inhibition rate. ****: *p* < 0.0001; ***: *p* < 0.001; **: *p* < 0.01; ns: *p* > 0.05.

## Data Availability

The authors declare that the data supporting the findings of this study are available within the paper and its Appendix A, or available from the corresponding authors upon reasonable request.

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
