# Peer review of "Four-Color Pseudovirus-Based Neutralization Assay: A Rapid Method for Evaluating Neutralizing Antibodies Against Quadrivalent Hand, Foot, and Mouth Disease Vaccine"

_vaccines, 2025, doi:10.3390/vaccines13030320_

Round 1
Reviewer 1 Report
Comments and Suggestions for Authors
This manuscript describes the development of reagents for the simultaneous measurement of infectivity and neutralization activity of four different picornaviruses that cause hand, foot, and mouth disease. This simultaneous measurement would make these assessments of neutralization activity far more efficient.
In general the experiments are straightforward and this is certainly valuable work. However, the manuscript omits or is unclear about a number of details in the experiments: these defects should be corrected.
1.What does it mean that (line 122) “P1 packaging sequences…were codon optimized using the CMV promoter system”? What is the relationship of the promoter to the codon optimization? And a few words (at least) about “packaging sequences” would be helpful for those of us studying unrelated viruses.
2.Line 144 mentions the CA16 replicon “as previously described”. Where can the reader find this “previous description”?
3.Line 150 states “the supernatant was harvested through a single freeze-thaw cycle”. I do not really understand what this means, and of course it is not a sufficient description to enable a reader to replicate the experiments. I would urge the authors to provide considerably more detail here.
4.Table S4 contains the “theoretical value” of NT50. What is this? I could not find any explanation of this parameter or how it was calculated.
A minor point: line 107 mentions “five” fluorescent reporter genes, but I only see four.
Author Response
Comments 1. What does it mean that (line 122) “P1 packaging sequences…were codon optimized using the CMV promoter system”? What is the relationship of the promoter to the codon optimization? And a few words (at least) about “packaging sequences” would be helpful for those of us studying unrelated viruses.
Response 1. Thank you for pointing this out. We agree with this comment. Therefore, we revised the sentence at line 123-125, which now reads: “The P1 sequences encoding capsid proteins of EV71, CA16, CA10, and CA6 were codon optimized (GENERAL BIOL, China) for human cells, cloned into the vector backbone, and expressed under the control of the CMV promoter.” Additionally, we have included Reference 22 to support this revision.
Reference 22: Choi, W.-S.; Oh, S.; Antigua, K.J.C.; Jeong, J.H.; Kim, B.K.; Yun, Y.S.; Kang, D.H.; Min, S.C.; Lim, B.-K.; Kim, W.S.; et al. Development of a universal cloning system for reverse genetics of human enteroviruses. Microbiol Spectr 2023, 11, e0316722. DOI:10.1128/spectrum.03167-22.
Comments 2.Line 144 mentions the CA16 replicon “as previously described”. Where can the reader find this “previous description”?
Response 2: Thank you for pointing this out. We agree with this comment. Therefore, we revised the sentence at line 145-147, which now reads: “For the CA6 replicon, we used the CA16 replicon with the p1 gene deleted and replaced by the RFP fluorescent reporter gene.” as shown in Figure 1E.
Comments 3. Line 150 states “the supernatant was harvested through a single freeze-thaw cycle”. I do not really understand what this means, and of course it is not a sufficient description to enable a reader to replicate the experiments. I would urge the authors to provide considerably more detail here.
Response 3. Thank you for pointing this out. Cell cultures that were transfected were frozen at -80 °C and then thawed in a room temperature water bath. we revised the sentence at line 158, which now reads: “ Following a 48-hour transfection period, the supernatant was harvested through a single freeze-thaw cycle (from -80℃ to room temperature) and centrifugation at 4000 × g for 20 min.”
Comments 4. Table S4 contains the “theoretical value” of NT50. What is this? I could not find any explanation of this parameter or how it was calculated.
Response 4. Thank you for pointing this out. We agree with this comment.
The theoretical value is the expected NT50 result following the appropriate dilution of a known NT50 antiserum. The theoretical value is used for comparative analysis with the actual test results to evaluate the accuracy of the analytical method.
To help explain it more clearly, we have included notes on theoretical values in Table S4: “The theoretical value is the expected NT50 result following the appropriate dilution of a known NT50 antiserum.”
Comments 5. A minor point: line 107 mentions “five” fluorescent reporter genes, but I only see four.
Response 5. Thank you for pointing this out. We agree with this comment. we revised the sentence at line 107-108, which now reads: “Following transfection of the expression plasmids for these four fluorescent reporter genes into 293T cells…”
Reviewer 2 Report
Comments and Suggestions for Authors
It is not possible to judge the results as the figures are too small to see anything. Figure 1 which really is critical to further review is useless at this point. In order to be reviewed it needs to be shown better.
Author Response
Comments 1. It is not possible to judge the results as the figures are too small to see anything. Figure 1 which really is critical to further review is useless at this point. In order to be reviewed it needs to be shown better.
Response 1. Thank you for pointing this out. We agree with this comment. We have increased the font size to enhance readability. Additionally, we fine-tuned the sharpness of Figure 1 so that the details of the spots appear clearer and more distinct.
Reviewer 3 Report
Comments and Suggestions for Authors
Fan Gao et al set up a four-color pseudovirus-based neutralization assay, in order to evaluate neutralizing antibodies against a quadrivalent HFMD vaccine. The method developed is rapid and simultaneously detects NtAbs against the four serotypes, requiring lesser amounts of sera compared to the gold standard technique, cell based neutralization assays. The topic of the manuscript is of interest and provides interesting results. The content is well structured and clearly presents the information in a meaningful way to the reader, which makes it easy to read. In addition, the manuscript follows a logical order, the results are clearly presented and the conclusions are supported by them.
However, I found some issues that should be addressed and are detailed below:
1- Lines 122-3: “The P1 packaging sequences of EV71, CA16, CA10, and CA6 were codon optimized using the CMV promoter system.” Please, provide a citation.
2- Lines 223-4: “RD cells were found to be the most susceptible to EV71, CA16, CA10, and
CA6 pseudoviruses (Figure 2A)” Did you find statistical differences in susceptibility among cell lines tested?
3- Lines 224-5: “The optimal testing time point was determined to be between 19 and 21 h post-infection (Figure 2B)” Please, explain briefly the basis of your selection.
4- Lines 225-6: “with an ideal pseudovirus addition amount of 500 pfu/well. The optimal cell inoculation density was 4–6 × 104 cells/well.” Please, include Figure 2C and D on those lines.
5- Figure 2D: on the references of each panel it says PFU/well but the legend says
5- Table S1: * indicates a statistically significant difference in the results when using P31.
What did you compare in order to obtain that difference? Which statistical analysis did you use? What was the p value?
6- Lines 254-6: “the cross-NtAb titers of anti-CA16 serum detected using the EV71 authentic virus can reach 6.25%, whereas our study yielded a titer of 5.29%”. Could you explain what those percentages stand for?
7- Table S4: How did you calculate the theoretical value of PBNA NT50?
Author Response
Comments 1. Lines 122-3: “The P1 packaging sequences of EV71, CA16, CA10, and CA6 were codon optimized using the CMV promoter system.” Please, provide a citation.
Response 1. Thank you for pointing this out. We agree with this comment. Therefore, we revised the sentence at line 123-125, which now reads: “The P1 sequences encoding capsid proteins of EV71, CA16, CA10, and CA6 were codon optimized (GENERAL BIOL, China) for human cells, cloned into the vector backbone, and expressed under the control of the CMV promoter.” Additionally, we have included Reference 22 to support this revision.
Reference 22: Choi, W.-S.; Oh, S.; Antigua, K.J.C.; Jeong, J.H.; Kim, B.K.; Yun, Y.S.; Kang, D.H.; Min, S.C.; Lim, B.-K.; Kim, W.S.; et al. Development of a universal cloning system for reverse genetics of human enteroviruses. Microbiol Spectr 2023, 11, e0316722. DOI:10.1128/spectrum.03167-22.
Comments 2. Lines 223-4: “RD cells were found to be the most susceptible to EV71, CA16, CA10, and CA6 pseudoviruses (Figure 2A)” Did you find statistical differences in susceptibility among cell lines tested?
Response 2. Thank you for pointing this out. We have incorporated statistical analysis into Figure 2A. During this process, we reviewed the original data and subsequently updated the figure:
Comments 3. Lines 224-5: “The optimal testing time point was determined to be between 19 and 21 h post-infection (Figure 2B)” Please, explain briefly the basis of your selection.
Response 3. Thank you for pointing this out. We calculated the coefficient of variation (CV) for each time point, selecting those where the CV value was below 30%. Among all time points that met this criterion, the virus titer reached a plateau between 17 and 24 hours. To ensure greater robustness of the method, we selected 19 to 21 hours.
Comments 4. Lines 225-6: “with an ideal pseudovirus addition amount of 500 pfu/well. The optimal cell inoculation density was 4–6 × 104 cells/well.” Please, include Figure 2C and D on those lines.
Response 4. Thank you for pointing this out. We agree with this comment. We revised the sentence at line 227-228, which now reads: “…amount of 500 pfu/well (Figure 2C). The optimal cell inoculation density was 4–6 × 104 cells/well (Figure 2D).”
Comments 5. Figure 2D: on the references of each panel it says PFU/well but the legend says
Response 5. Thank you for pointing this out. We agree with this comment. We revised Figure 2D:
Comments 6. Table S1: * indicates a statistically significant difference in the results when using P31. What did you compare in order to obtain that difference? Which statistical analysis did you use? What was the p value?
Response 6. Thank you for pointing this out. I conducted a comparative analysis of the NT50 results for CA10 on the P3, P14, and P31 generations of RD cells. An unpaired student’s t-test was employed to evaluate the differences between the P14 and P31 results, which revealed a statistically significant difference (P=0.0411).
To help explain it more clearly, we revised the notes in Table S1: “* indicates a statistically significant difference in the results between P14 and P31 (P=0.0411).”
Comments 7. Lines 254-6: “the cross-NtAb titers of anti-CA16 serum detected using the EV71 authentic virus can reach 6.25%, whereas our study yielded a titer of 5.29%”. Could you explain what those percentages stand for?
Response 7. Thank you for pointing this out. This percentage represents the ratio of the CA16 neutralizing antibody titer to the EV71 neutralizing antibody titer, expressed as a percentage. Additionally, we have included Reference 27 to support this revision.
Reference 27. Cai, Y.-C.; Ku, Z.-Q.; Liu, Q.-W.; Leng, Q.-B.; Huang, Z. A combination vaccine comprising of inactivated enterovirus 71 and coxsackievirus A16 elicits balanced protective immunity against both viruses. Vaccine 2014, 32, 2406–2412. DOI: 10.1016/j.vaccine.2014.03.012.
Comments 8. Table S4: How did you calculate the theoretical value of PBNA NT50?
Response 8. Thank you for pointing this out. We agree with this comment.
The theoretical value is the expected NT50 result following the appropriate dilution of a known NT50 antiserum. The theoretical value is used for comparative analysis with the actual test results to evaluate the accuracy of the analytical method.
To help explain it more clearly, we have included notes on theoretical values in Table S4: “The theoretical value is the expected NT50 result following the appropriate dilution of a known NT50 antiserum.”
Round 2
Reviewer 2 Report
Comments and Suggestions for Authors
My primary concern with the first draft was that section G in Figure 1 could not be judged. The authors have improved the figure although i still think it could be brighter. However, if other reviewers are convince then i am ok with the present version.
Reviewer 3 Report
Comments and Suggestions for Authors
Authors have done a great job at modifying the manuscript, the revised version is considerably clearer than the old one. In addition, the authors have answered in detail all of my queries, thank you very much.